# Degradation Processes of Medieval and Renaissance Glazed Ceramics

**DOI:** 10.3390/ma16010375

**Published:** 2022-12-30

**Authors:** Mária Kolářová, Alexandra Kloužková, Martina Kohoutková, Jaroslav Kloužek, Pavla Dvořáková

**Affiliations:** 1Department of Glass and Ceramics, University of Chemistry and Technology Prague, Technická 5, 16628 Prague, Czech Republic; 2Central Laboratories, University of Chemistry and Technology Prague, Technická 5, 16628 Prague, Czech Republic; 3Laboratory of Inorganic Materials, University of Chemistry and Technology Prague, Technická 5, 16628 Prague, Czech Republic; 4Institute of Rock Structure and Mechanics ASCR, Czech Academy of Sciences, V Holešovičkách 41, 18209 Prague, Czech Republic

**Keywords:** archaeological lead-glazed ceramics, archaeometric characterisation, degradation, glaze fit

## Abstract

Corrosion effects in deposit environments (soil, waste pit, etc.), together with the glaze adherence and fit, could cause severe deterioration accompanied by different types of defects or growth of corrosion products. The aim of this work was to identify the source of surface degradation of the lead-glazed ceramics sets from the Prague area from the Romanesque to the Renaissance period. A combination of X-ray fluorescence (XRF), X-ray diffraction (XRD), optical microscopy (OM), scanning electron microscopy with energy dispersive X-ray spectroscopy (SEM/EDS), and simultaneous thermal analysis (STA) techniques along with stress state calculations was used to study the defects. Based on the interpretation of the possible sources of the observed defects, four types of degradation effects were schematically expressed for the archaeological samples. It was shown that the glazes were already appropriately chosen during the production of the Romanesque tiles and that their degradation occurred only due to long-term exposure to unsuitable environmental conditions.

## 1. Introduction

The majority of historical glazed objects, whether they are utility goods, technical ceramics, stoves, or ceramic tiles, were provided with lead glazes with various colouring additives [1,2,3,4,5,6]. Their use was associated with a simple method of production (application by pouring and a relatively low firing temperature) and the achievement of the required aesthetic parameters (mainly gloss and rich colour shades). Due to the influence of the surrounding environment during the use and subsequent storage of ceramics in the earth, degradation of both components—glazes and shards—took place. In the soil or pit environment, the main causes of degradation are the inappropriate humidity and temperature values or the concentration of inorganic and organic agents (pH of the environment); the greatest damage of the findings is mainly caused by the fluctuation of these corrosion parameters [7,8,9,10,11]. The compatibility of the two-component system of the glaze and the substrate-chipped matter has a fundamental influence on the long-term stability of the decorative layers. The compatibility can be influenced both by the composition of the two components and by the production conditions, especially the firing (e.g., the density of the glaze suspension during the application, the viscosity of the glaze during firing, the reactions of the glaze with the kiln atmosphere, the temperature profile of the firing, the formation of an intermediate layer between the glaze and the shard, diffusion, and sintering between the glaze layer and the shard material, etc.) [1,12,13,14,15,16]. When evaluating the compatibility of the shard material and the glaze (glaze adherence, glaze fit), it is necessary to monitor the extent and nature of the stresses that arise in the glaze and in the shard during cooling in the final phase of firing [17]. Glazes are generally poorly resistant to tensile stress; on the contrary, they withstand high-pressure loads up to approx. 100 MPa without developing defects [17]. When a certain strength limit of the glaze is exceeded, defects of various types occur depending on the nature of the stress. In the case of tensile stress, it is the crazing or cracking of the glaze; in the case of compressive stress, the glaze peels off from the surface [18,19,20,21,22]. Individual types of malfunctions appear either immediately after firing or during the use or storage of the objects in the soil.

The defectoscopy (defect control) of glazes can be divided into two areas:Study of defects created by the interaction of the glaze with the shard;Study of surface defects of glazes caused by the effect of the surrounding environment on the glaze layer itself.

Both areas can simultaneously contribute to the formation and growth of defects, i.e., the mismatch between the stress states of the glaze and the shards, as well as the action of corrosion factors. The first step in identifying the sources of defects is the evaluation of the consistency of the glaze and the shard material, followed by the evaluation of the degradation processes associated with the identification of the corrosion products. The chemical stability of the glaze layer can be influenced by three types of factors, namely the surrounding environment (temperature, relative humidity, exposure time, the presence of pollutants or microorganisms, etc.), the physical conditions of the degradation (e.g., dynamic or static conditions, S/V (surface/volume) ratio or area/volume, etc.) and the condition of the material that is subject to degradation (thermal history, surface condition, homogeneity, composition, etc.) [10,16]. Contamination from the surrounding environment can occur during storage (e.g., in soil), during inappropriate treatment (during restoration work), as a result of specific use (e.g., storage of goods from which corrosive media are released), or placement in an environment with a higher concentration of acidic substances (waste pits) [23]. The complex mechanism of glaze corrosion can be expressed as the study of the main reactions of glass materials in an aqueous environment with different pH levels. Depending on the amount of water condensed on the surface, the given process can be characterised as corrosion by air or corrosion by solutions. Changes in the temperature and humidity of the surrounding environment can lead to the precipitation of a layer of products of a microporous structure on the surface of the glaze, and with further growth of this corrosion layer to the peeling of the corrosion crusts [24]. The process of dissolution in a liquid environment has been studied for different types of glass [25,26,27,28,29,30,31,32], where several parameters have been analysed, e.g., reaction medium, the chemical composition of the glass, temperature, etc. Even though ceramic glazes are very similar to glasses, the study of the process of dissolution and overall corrosion is more complicated due to the high number of components in the glaze raw materials. However, the study of glass corrosion can be a guide for a comprehensive description and prediction of glaze degradation.

The experimental determination of the stress relations and behaviour of glazed ceramic materials has an essential importance for evaluating the development of defects and deformations induced by the state of the stress established in the glaze-ceramic substrate system. The theoretical determination of flexion behaviour and stress state of glazed ceramic materials has considerable limits. The commonly used experimental methods to evaluate the stress or glaze to ceramic body adhesion are mainly the Steger method, glazed ring method, or different types of thermal analyses, i.e., optical dilatometry, dilatometry, or thermomechanical analyses [33,34,35,36,37,38]. A comparison of the measured curves and the coefficients of thermal expansion (CTE), the transformation temperatures (T_g_), the set points (T_n_), and other characteristic temperatures, i.e., the softening point (T_d_), could be used to evaluate the stage of the stress (σ) developed within the glaze and its substrate in MPa. The results of the graphical comparison of the measured curves, which are shifted to the set point of the glaze, and the results of the differences in the coefficients of linear thermal expansion can be compared with the calculations of the stress relations [16,39,40,41,42,43,44,45]. Recent studies on the glass proved that the use of differential scanning calorimetry (DSC) or differential thermal analysis (DTA) is also suitable for obtaining the characteristic temperature, such as the glass transition temperature T_g_, onset crystallisation temperature T_x_ crystallisation temperature Tc and melting temperature T_m_ [44,45,46,47,48].

The aim of the presented work is the defectoscopy of lead glazes of archaeological finds with the determination of the source of their damage. The collection of glazed ceramic objects and fragments was selected to capture various types of Prague finds from the late Romanesque period (11th century) to the High Renaissance (18th century) [49,50]. All the objects were collected as part of rescue archaeological research in the last four decades in the area of the Benedictine Monastery (Břevnov Monastery), the grounds of the Prague Castle, and other houses and palaces in Prague (Figure 1). The aim of this study was to determine whether degradation is the cause of primary (instant) effects or the result of secondary (delayed) action of the surrounding. Defects in ceramic products can appear during any stage of production and can be related to many simultaneous causes. The goal of this work was focused on individual defects, which were distinguished according to the source of their formation and according to their characteristic form.

## 2. Materials and Methods

### 2.1. Studied Archaeological Glazed Ceramics

For the study of the degradation processes of glazes, fragments of archaeological finds of various types and dates, from the Romanesque period to the Gothic period to the Renaissance, were selected. The evaluated set contained more than 850 fragments, and 24 glazed samples were used to investigate the identification of the origin and cause of the defects (Table 1): four fragments of Romanesque (R1, R2, R3, and R4) and three Gothic (G1, G2, and G3) glazed tiles (floor tiles), eight fragments of Renaissance tiles (S1–S8 stove tiles), two fragments of relief large-format tiles (W1 and W2 relief wall tiles), three samples of technical ceramics (T1, T2, and T3) and the last set contained utility ceramics, which includes a representative of kitchen goods (M1) and two faience imports (F1 and F2).

### 2.2. Initial Characterisation of the Glazes and Their Ceramic Bodies

The initial characterisation of the chemical and mineralogical composition of the glazes and ceramic bodies of the studied set was performed using X-ray techniques. The chemical compositions of the ceramic bodies in the form of powdered samples and glazes in the form of flat samples were determined by X-ray fluorescence analysis (XRF). A fully automatic sequential WD-XRF Performix spectrometer (Thermo Fisher Scientific, Switzerland) equipped with an Rd anode X-ray tube was used for the analysis. The acquired data were evaluated using Oxsas standardless software (Thermo Fisher Scientific, Switzerland), and the measured data were further evaluated based on the measurement error. The chemical composition of the ceramic bodies and glazes was summarised by a principal component analysis (PCA) using XLStat statistical software for MS Excel. This statistical classification divided the obtained data according to quantitative variables, i.e., the identified oxides. The PCA method is used as a statistical evaluation of data obtained from XRF analyses for large sets of shards or input raw materials for their illustrative and effective differentiation [51,52,53]. The PCA analysis efficiently divides data files into individual clusters. Thanks to the possibility of incorporating all the variables (including the minority phases) into the calculations, it provides an effective and illustrative comparison of the composition of a large data set. Depending on the composition, the principal component analysis method could indicate the tendency to corrode in a certain environment (e.g., change in the alkali content due to an alkaline environment, change in the ratio of the lead component due to an acidic environment, etc.). The corrosion products observed on several glazed surfaces were also evaluated by XRF analysis. Due to the small amount and problematic sampling of the corrosion products, an Axios PANanalytical sequential XRF spectrometer enabling the measurement of a 6 mm area in diameter was used. The data were evaluated using the standardless software Omnian. The mineralogical compositions were identified by X-ray diffraction analysis (XRD) using a PANanalytical X’Pert PRO θ-θ powder diffractometer (PANanalytical, Holland). The ceramic bodies and corrosion crusts were analysed in the form of fine powders, and all the glazes were analysed after chemical cleaning with a 5% acid solution from the flat surfaces. The X-ray diffraction data were measured at room temperature using CuK_α_ radiation over the angular range of 5–80° 2θ. The measured data were evaluated using the PANalytical High Score Plus 4.0 software package (PANanalytical, Holland). The semi-quantitative analysis of the ceramic bodies and glazes was performed using the Reference Intensity Ratio (RIR) values from the PDF4+ database.

### 2.3. Evaluation of the Glaze Fit

The thermal analyses, especially the dilatometric measurements (DIL), thermomechanical analysis (TMA), and simultaneous Thermogravimetry/Differential Thermal Analysis (TG-DTA), were used to evaluate the glaze characteristic temperatures, glaze fit, and glaze adherence. Due to the limited amount of glaze samples of some of the studied archaeological fragments, the sample properties were calculated as a function of the composition using the Priven-2000 empirical model of SciGlass software and the glass property database [54,55]. The validation of usability of the software was previously verified on the medium and high-lead models and archaeological glazes [17].

For the simultaneous thermal analysis (Linseis Messgeraete GmbH, Selb, Germany)) in the TG-DTA mode, the LINSEIS STA Platinum Series 1600/1750 °C HiRes system for 50 ± 0.05 mg of a powdered sample was used. The measurement was carried out in a platinum crucible with a lid at the heating rate of 10 °C min^−1^ or 5 °C·min^−1^ in a temperature range from room temperature (RT) to 800, 900, 1000, or 1200 °C. A controlled helium flow rate of 20 mL min^−1^, to ensure an inert atmosphere during the measurement, was used. The gas release (mainly H_2_O and CO_2_) was measured through a silica capillary with a ThermoStar^TM^ GSD320 quadrupole-type mass spectrometer (Pfeiffer Vacuum GmbH, Aßlar, Germany) in the range of 300 AMU. The Linseis Acquisition software controlled the measurement, and the detection of released gases was performed using the Quadera 4.62 software. The measured data were evaluated in the Linseis TA Evaluation software. The dilatometric measurements of ceramic bodies and glazes were carried out using a LINSEIS L75 HS 1600C PT (Linseis Messgeraete GmbH, Selb, Germany) dilatometer. The ceramic bodies were analysed in the form of compact rectangular prism samples (dimensions of approx. 20 × 5 × 5 mm), and the glazes were analysed in the form of cylindrical samples (size of approx. 20 × 5 mm), which were cut from the fired compact samples. The measurements were carried out at heating rate of 5 °C min^−1^ and with a zero adjustment at the controlled force of 300 mN in a helium flow. The temperature range was 25–700 °C for the ceramic body samples and 25 °C for the deformation temperature of the glaze samples. The measuring process of the ceramic body samples consisted of several stages. First, annealing at a selected temperature mode was performed with dried compact samples. The strain-temperature dependence from the first annealing is influenced by thermal expansion and also by contraction due to the dehydroxylation (elimination of hydroxyl groups). The second measurement of the sample was performed on the previously analysed compact sample, and the degree of rehydroxylation of the sample was zero. The difference between these two annealing steps is the degree of natural irreversible moisture expansion of the rehydroxylation of the annealed sample. The second annealing dependence, therefore, corresponds to the thermal expansion of the studied ceramic body. The third and fourth measurements were performed only for selected compact ceramic body samples that were subjected to hydrothermal treatment in the autoclave (180 °C, 1 MPa, 5 h) [56,57,58]. The difference between the second and third measurements is identical to the maximum irreversible expansion due to the accelerating test. The fourth measurement verified the second measurement was affected only by the thermal expansion.

### 2.4. Identification of the Origin of Deterioration

Surface conditions, types of defects, and the presence of corrosion layers or products were examined using microscopic observations. An Olympus BX51 polarising optical microscope (Olympus, Japan) with or without crossed polarisers and a Canon 500D digital SLR camera were used to investigate thin sections of the selected archaeological samples. An Olympus SZ61 stereomicroscope (Olympus, Japan) with a ProgRes CT3 digital camera was mainly used to study the surface conditions and defects of the individual fragments. The images were processed in NIS Elements AR 4.60 and ZoomBrowser EX software. The obtained data were supplemented by evaluating the homogeneity, corrosion layers, pigments, and glaze morphology of the crystalline phases using a Tescan VEGA 3 LMU electron microscope (Tescan Orsay Holding, Czech Republic) equipped with a chemical microanalysis using an OXFORD Instruments INCA 350 Energy-dispersive X-ray spectroscopy (EDS) analyser.

## 3. Results

### 3.1. Initial Characterisation of the Studied Samples

The initial characterisation of the chemical composition of shards and glazes was performed using the XRF method, and the measurement results were statistically evaluated using principal component analysis. Figure 2 and Figure 3 show the results of the PCA analysis of the fragment masses, respectively glazes, and their division into colour groups according to the type of product.

The results show that the variability in the composition of the shards is primarily determined by the typology of the ceramic product. The sets of the oldest finds of Romanesque and Gothic tiles (brown circle) show the greatest variation in the chemical composition of the shard materials; the composition is more variable in the younger group of finds. The statistical evaluation shows that the shards are similar and differ primarily in the mutual ratio of silicon dioxide and aluminium, where the ratio is the highest in the oldest shards. A higher content of titanium dioxide was recorded in the younger finds, and a higher proportion of mainly Fe_2_O_3_ and minor components of MnO were recorded in the fragmentary materials of the tiles. From the point of view of comparing the differences according to the different use of individual shard materials, the shard materials of the technical ceramics are closest to the stove tiles.

The chemical composition of glazes is correlated in Figure 3, from which can be seen that the individual glazes can be divided mainly based on the presence of different types of colouring additives and the content of the lead component, which are medium to high lead glazes with a lead oxide content in a range of 14–58 wt.%. This variability in the PbO content can be caused by high inhomogeneity (especially in the glaze samples of the oldest date) as well as by the dissolution of the Pb component during storage in the ground. The individual glazes are divided according to the different colouring components present, mainly in the form of ionic dyes, Fe, Cu, Mn, Sb, Co, and lime in the form of SnO_2_ or other minor components, e.g., ZnO. In terms of the phase composition, the amorphous phase significantly prevailed in all of them; the crystalline components mainly contained lead feldspars with variable composition (lead aluminosilicates with an admixture of alkaline cations), colouring admixtures, potash, or corrosion products. The highest proportion of Fe_2_O_3_, similar to that of the shards, was found in the glazes of the Romanesque and Gothic tiles, R1–4 and G1–3, respectively. In the Gothic tiles G1, the colouring component in the form of crystalline hematite was confirmed (XRD), and individual clusters of crystals were observed by light microscopy (Figure 4). Various Fe-minerals were found in the R1, R2, R4, and G3 tiles, as well as hercynite (FeAl_2_O_4_) in the R4 tile, Fe-mineral with Pb admixture in the R1 and G3 tiles, another admixture could be fayalite (Fe_2_SiO_4_). For all the tiles, the corrosion products in the form of hydrated ferric and lead phosphates and lead feldspar formed by firing were identified (XRD).

The majority of the brown-coloured glazes (R1–R4, G1–G3, S1–S6, and M1, see Table 1) also had varying proportions of Mn (0.3–1.6 wt.%) or lighter shades of blue and green-brown glazes of tiles S3 and S4 also had proportions of the ionic dye based on Co (4–5 wt.%) and the light brown shade glazes of samples S1, S4, S5, and S6 were influenced due to the presence of Sb_2_O_3_, which can act in low-melting glazes like a slurp. All the green glazes of tiles S8 and S9 and the large-format wall reliefs W1 and W2 contained colouring components with a combination of Cu (0.2–4 wt.%) and Fe (0.3–1 wt.%) oxides. This combination of ionic dyes was also the main colouring admixture of other Renaissance green glazes on the technical ceramics T1, T2, and T3 and the youngest bowl find presented. They were combined with the melting admixture ZnO, which positively affects the resulting gloss of the glaze, and the opacifier SnO_2_ was confirmed in the form of cassiterite (XRD). Cassiterite was also identified as the main crystalline phase in the glazes of the faience imports F1 and F2, whose white and blue glazes contained 3.4–11 wt.% SnO_2_.

In the white and blue glazes of samples F1 (Figure 5), and in addition to cassiterite, the presence of a mineral from the marcasite group was also recorded, which is an orthorhombic polymorph of isometric pyrite, or a group of common disulphides, was also noted. The presence of this minor component can be caused by chemical, biological damage (chemical biodeterioration of transition metal oxides which are represented as colourants in glazes) of the transition metal oxides, which act as colourants in glazes. This type of damage to archaeological ceramics with high-lead glazes is also referred to as blackening caused by sulphur-reducing bacteria [59]. In the white faience glazes applied to a shard with a high proportion of CaO and MgO, the reaction with the underlying shard at the interface preferentially produces calcium silicate-based minerals instead of lead feldspars. The presence of S (0.3–6 wt.%), P (in the form of P_2_O_5_ in the range of 0.8–5 wt.%), and Cl (0.4–1.4 wt.%) was recorded for most of the glazed finds from the pit environment. Mainly quartz, K-feldspar, and Ca-albite were identified in the fragmentary materials, which came from the colouring components hematite (tile), anatase, or rutile. Furthermore, a higher proportion of CaO was identified in the shards as anorthite or gehlenite, or diopside (faience imports, stove tiles S7, S8, and W1). In addition to the main minerals, mullite was also identified in the basin (M1).

### 3.2. Evaluation of the Glaze Fit and Identification of the Causes and Origins of Defects

The individual groups were evaluated, which were sorted by age and defects. First, a set of Romanesque tiles was presented, then tiles from Gothic mosaic paving and modern utility goods (pan) were studied, followed by the most numerous set of Renaissance tiles; the last group consists of representatives, faience imports, and technical ceramics and relief large-format tiles.

In the first set of Romanesque glazed tiles, numerous cracks and surface corrosion crusts were identified during the initial survey. To determine possible sources of the damage, calculations of the stress states of the glaze-ceramic shard systems were performed. The values of the longitudinal expansion of the shards were obtained dilatometrically, the values of the R1 and R4 glazes were measured with a thermomechanical analyser and compared with the calculations by the SciGlass software, and for the R2 and R3 glazes, the data calculated by the software were used to calculate the stress states. Corrosion products contain mainly phosphorus, lead, and calcium in the form of hydrated lead-calcium phosphates, e.g., wicksite, or carbonates, mainly cerussite. Furthermore, various degrees of surface damage were confirmed by the microscopic observations (Figure 6a–c).

The course of the stress states of the samples of the Romanesque tiles calculated from the second dilatation measurements of the fragments characterise the state of the samples after firing (Figure 6d,e). It was proven that the glaze-shard mass ceramic body systems were in accordance immediately after firing or the glazes were in moderate compressive stress. The total differences in the stress states Δ (total mismatch) between the glaze and the shard are in the range of 0.84–66.19 MPa. The limit value for peeling is approx. 100 MPa, i.e., no damage was caused by exceeding the strength limit in any of the cases (Table 2). Conversely, the compressive stress in the glazes (from −0.05 MPa for R1 glaze to −2.54 MPa for R3 glaze) led to an increase in the resistance of the surface decor to cracking, and even a very humid environment during storage did not lead to tensile cracks. The cracks observed on the surface (Figure 6a–c) were formed as a result of the crystallisation and growth of corrosion crusts during the long-term storage in the moist soil near the skeletal remains. The corrosion crusts mainly contained Ca and Pb (6bc), which correspond to the identified cerussite PbCO_3_. P, Ca, and Pb (carbonates and hydrated calcium and lead phosphates) were identified in the area of the crust edges. The corrosion crusts grew through the glaze until they came into contact with the substrate, and their expansion caused the glaze to peel off. When cracks from the centre of the corrosion crusts developed, a local change in the stress and a violation of the strength limit of the glazes in the place of the crusts took place. Even though the paving was made with the glaze in a moderate compressive prestress (i.e., in line with the substrate), it significantly degraded when placed in the ground. In the relief-decorated tiles (mainly R1 and R4), the plastic decor was damaged by surface mechanical abrasion, visible in Figure 6b. This type of defect was mostly observed in tiles located in exposed areas, where surface abrasion was visible to the naked eye and occurred during the use of the rotunda.

The defectoscopy of the glaze of the younger Gothic group of the archaeological paving G1 was evaluated using optical analyses. On the surface of the Gothic tile samples, only the partially preserved remains of the glaze were strongly cracked. A tiny interlayer between the glaze and the porous sherd mass was identified as peeling, see Figure 7a. Its positive effect did not manifest in the mechanical strength of the glaze. The release of tension between the glaze and the shard manifested in the chipping of the glaze. The optical microscope image in Figure 7a shows a relatively large grain of the shard penetrating from the shard mass up to the surface into the glaze layer. The grains (mainly quartz and potassium feldspars) raised in this way were captured in all the samples of the Gothic tiles and could negatively affect the cohesion of the entire system. This is evident from the crack that developed near the penetrating isometric quartz grain in Figure 7a,c. Due to the low preservation of the glazes on the surface of the Gothic tiles, it was not possible to prepare samples for the TMA measurements. The data calculated by the SciGlass software from the chemical composition were used to calculate the stress relations of the systems. It was proven (Figure 7b,d) that the main cause of the loss of the glazes of the Gothic tiles was the inconsistency of the given system and the extreme stress (more than 100 MPa), which, together with the unevenness of the glaze layer, a relatively large number of unreleased bubbles in the glaze and large grains of grit at the interface between the glaze and the substrate (with high porosity) led to the total destruction of the surface.

In the case of the fragments of the M1 pan, a network of fine cracks was observed (Figure 8a,c); on some of them, which were deposited in the same layer of the waste pit, signs of degradation in the form of corrosion pinholes (chemical alteration with the environment, corrosion pinholes) were captured. On one fragment, parallel grooves were visible on the surface layer (they could have been caused mechanically during use or even during storage in the pit environment), which helped the corrosive effect of the environment. Although the total difference in the stress states in the glaze-substrate system in the M1 pan samples was 125.35 MPa, there was no damage due to the tensile stress in the glaze, the compact highly amorphous layer of glaze with a thickness of approx. 200 μm and a more pronounced intermediate layer at the interface between the glaze and the shard, no peeling occurred in the pan sample, which could be expected due to the high value of the total stress.

Based on the measured values from the dilatometer and the thermomechanical analyser, tensile stress leading to cracking of the glazes was detected in the tile glazes. The measured relative expansions were compared with each other and then used to calculate the stress states (Figure 9d) and verify the applicability of the data calculated in the SciGlass software (Figure 9b). During the characterisation of the surface defects of the tile fragments, corrosion pits, which were formed on the surface secondary to the action of corrosive media in the pit, were captured in Figure 9a,c next to the tensile cracks corresponding to the stress state of the system. In the layers of the pit, from the environment of which most of the fragments of tiles with a corroded surface were lifted, an increased content of oxides of phosphorus (21 wt.% P_2_O_5_) and sulphur (7 wt.% SO_3_) was identified by the XRF analysis. Sulphates were also determined in the clays of the individual layers of the pit in the form of leachates in values of up to 5563 mgkg^−1^, which caused the activation of corrosion and the formation of defects. From the longitudinal expansion data calculated by the software and measured by the thermomechanical analyser in Figure 9b, it emerged that by including the overall chemical composition, including the minor components, in the calculations of high and medium lead glazes with a high proportion of amorphous phase, the measured data for the study of the development of stress relations can be replaced very easily.

The two three-component systems (shard-glaze-underglaze decor) of the imported faience, which have a different chemical and mineralogical composition due to the higher content of calcium and magnesium components compared to the other studied samples, were also included in the selection on the study of the degradation processes and differed from each other in the content of the individual types of calcium minerals. Figure 10b,d shows the development of stress relations of both two-component systems, which must be evaluated separately in order to identify the stresses in the individual components of the system. Figure 10b captures the course of stress in the system white glaze-shard material and demonstrates the compressive stress in the white glazes of both samples F1 and F2 very well. The graph in Figure 10d shows a comparison of the stress of the white covering glaze and the underglaze blue decor, where the white glaze, in both cases, on the contrary, appears in tensile overstress and is, therefore, in line with its overglaze decoration, which must be in compressive overstress to maintain the consistency of the system. The optical microscope image of Figure 10a shows the high-quality manufacturing of these luxury imports. Unfortunately, even with highly processed glazed objects, degradation occurs due to the instability of the colouring components and the presence of a highly corrosive environment, which can be observed in Figure 10c, where the corrosion of the blue underglaze decor of sample F2 with a high proportion of colouring components based on Co^2+^ ions is visible.

The last group of studied samples of archaeological glazed ceramics is a set of technical ceramics and two large-format relief-decorated lunette tiles, in which microscopic observations showed similar manifestations of the degradation effect of green glazes (Figure 11a,c), which have a very similar chemical composition. In these samples, the presence of unstable non-crystalline remains of plastic components in the shard mass was demonstrated using STA (see Figure 11b). The extent of the metal expansion was confirmed by dilatometric measurements; during repeated dilatation tests of samples of sherd materials W1 and W2 and T1, T2, and T3, significant changes in the curves documenting the dehydroxylation of the shard were identified. Sample W2, with the highest identified share of natural moisture expansion (Figure 11d), was subjected to hydrothermal ageing in an autoclave (according to the NF P13-302 standard [60]). After hydrothermal ageing, the sample was again measured with a dilatometer, and the stress states were calculated from the data. It follows from the calculations that the shard mass changed the stress of the system due to moisture expansion, and the compressive stress in the glaze turned into tensile stress. The identified state of the system supported corrosion processes in the glaze and the formation of corrosion products on the surface. An increased concentration of sulphur (2.35–3.93 wt.%), chlorine (2.85–10.79 wt.%), and phosphorus (3.5–11.67 wt.%) was identified in the corrosion products by the chemical microanalysis. It can be assumed that these are corrosion products in the form of sulphates, chlorides, and phosphates, especially hydrated calcium phosphates, lead and copper sulphates, and chlorides. Overall, the results confirmed that, due to the presence of unstable non-crystalline phases (metaclay), the moisture expansion of the shards occurred already during use and subsequently during storage, which could promote corrosion activity on the surface of the glazes.

## 4. Discussion

To identify the stress states of the studied glaze-shard mass systems, calculations of the stress states and values of their total difference (total mismatch/total difference) were used, which were compared with a graphical evaluation of the measured expansion or thermomechanical curves. A summary of the measured data and calculated stress states, together with the most frequent defects and identified corrosion products, is presented in Table 2. Based on the obtained results of the studied group, four possible types of degradation of archaeological glazed ceramics were identified/determined, which are schematically shown in Figure 12.

Figure 12A shows two manifestations of the primary defects-peeling/flaking of the glaze layer (peeling) and crazing, which occur when the glaze is not compatible with the shard material, resulting in excessive stress in the system, which is released by the growth of defects. Examples of these types of primary defects are the G1–G3, S1–S8, and M1 systems. The failure of the glazes of tiles S1–S8 and pan M1 by a fine network of cracks was caused by tensile stress induced in the glaze layer, which has a higher coefficient of thermal expansion than the shard. For the peeling or chipping off the glaze layer, or even with a thin layer of the shard, then it was observed for shards of Gothic tiles G1–G3 due to exceeding the limit value of the compressive stress due to shrinkage during firing. The rate of exceeding the compressive stress is influenced by the composition of both components of the glaze-shard mass system and the firing conditions.

Another manifestation of the degradation of the glazed ceramics can be the gradual crazing (delayed crazing)—Figure 12B, which occurs due to the influence of moisture. This type of degradation was observed in the samples of the relief large-format tiles (W1 and W2 relief wall tiles) and the technical ceramics (T1, T2, and T3), in which slow crack propagation (opening of the crack front), and thus, a decrease in the material strength occurred due to the influence of moisture. A violation of the surface could also be supported by fluctuations in the temperature or other parameters of the surrounding environment. This type of defect, which developed during use or subsequent storage, was caused by processes acting in the shard (rehydration, rehydroxylation, moisture expansion) and the related changes in the glaze layers.

Figure 12C shows the cause of the formation of defects in the form of secondary peeling of glazes (“delayed” peeling). This type of degradation was observed in samples R1, R2, and R4; in the case of R4, it was accompanied by defects in the form of bubbles or speckling. An imperfectly prepared substrate with an uneven surface and large grains sharpens the admixture in contact with the glaze or the growth of corrosion crusts under the glaze where the sources of the resulting defects occur. In the shard mass, with a large number of unsorted grains of shard near the surface, stress occurred at the boundaries of these grains, which led to an increase in the compressive stress of the glaze above the strength limit. Another reason for the disrupted coherence of the system was the retained bubbles of a size close to the thickness of the glaze layer near the glaze surface or possibly the migration of soluble salts through a ceramic shard with high porosity and a high proportion of unstable non-crystalline phases. It was found that as the surface roughness increases, the strength of the glaze decreases due to the growth of surface defects. The increasing roughness also negatively affected the accumulation of dirt on the surface. Due to the growth of surface corrosion products, the strength failure limit of the R1 glaze was exceeded, which led to its peeling.

The last type of degradation process, captured in Figure 12D, is the cooperation of stress states together with the possible corrosive action of external agents, which negatively affect the harmony of the glaze-shard material system. The formation and growth of corrosion products contribute to the acceleration of the formation of defects, which preferentially precipitate in places of reduced cohesion and a disturbed surface, which can be observed in the technical ceramics T1 and the Romanesque tiles R1 and R3.

## 5. Conclusions

Archaeological glazed ceramic finds with porous shards show a number of defects. The aim of studying their surfaces was to describe the damage and determine the source of the damage. From the results of the defectoscopy of the glazes of a selected set of archaeological finds, four types of degradation of the glazed ceramics were traced. The knowledge gained is useful for both contemporary and historical glazed ceramics and applicable to a wide range of samples. The scheme in Figure 12 was formulated based on the obtained results and shows a schematic guide on how to proceed when detecting possible sources of faults in glazed ceramics. The mentioned procedure can be applied to any kind of ceramics and for the preparation and study of a large series of model samples and the search for the causes of their degradation. Stress relations have not yet been closely observed in the field of archaeological ceramics, even though degradation of this type of material is a very frequent, common phenomenon. It was found that early medieval ceramic workshops in the Czech environment managed the production process of the first glazes relatively well. Some of the studied samples (W, T) were shown to have well-chosen glazes for the given shard materials. In terms of coefficients of longitudinal expansion, these glaze-shard mass systems showed appropriate values, or the thermal expansion coefficients of glazes and shards were consistent and not a source of damage. The observed defects in the glazes of the archaeological finds were caused by external factors. The exceptions are the Renaissance stove tiles S1, S4, S5, S7, and S8, which were created in such a way that characteristic tensile cracking of the glazes occurred. As a result of the ageing of the shard material during long-term storage in a soil environment with higher humidity, the tensile stress in the glaze layer shifted to critical values. This led to a decrease in the cohesion of the two-component glaze-shard mass system and accelerated the corrosive action of the surrounding environment. The crystallisation and growth of the corrosion products preferentially took place in places on the disturbed surface, in a network of fine cracks, or pitting corrosion occurred. In the samples of the studied Gothic tiles, a primary defect resulting from inappropriate setting of the stress states was also recorded. Faience imports, as the only representative of the three-component system, were damaged mechanically during storage in the pit environment. Broken surfaces were the cause of the acceleration of the biodegradation processes. Future studies should investigate the following issues: to focus on the study of three or more component systems, which in this work were represented by only one sample of faience import, and on the identification of possible raw material sources for the preparation of glazes, which could be relatively complicated.

## Figures and Tables

**Figure 1 materials-16-00375-f001:**
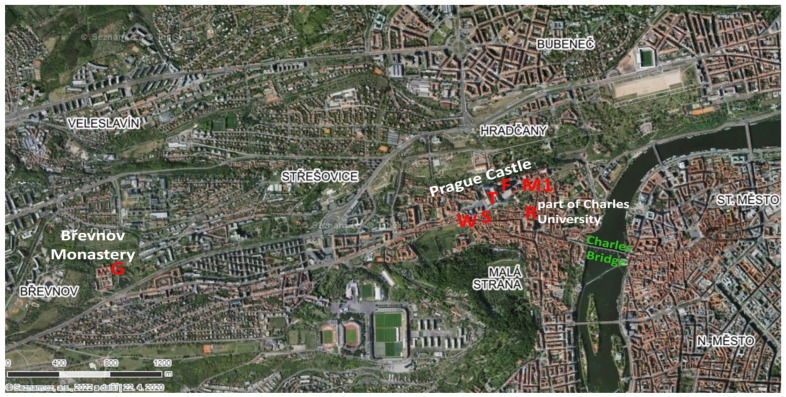
Map of the Prague district indicating the location of the studied sets: R—Romanesque St. Wenceslas Rotunda at Charles University, G—Gothic Benedictine Arch-Abbey, S—Salm Palace, W—Schwarzenberg Palace, T—Prague Castle complex waste pit B, F—Prague Castle complex waste pit H and C, M1—Prague Castle complex waste pit C. Source: Mapy.cz.

**Figure 2 materials-16-00375-f002:**
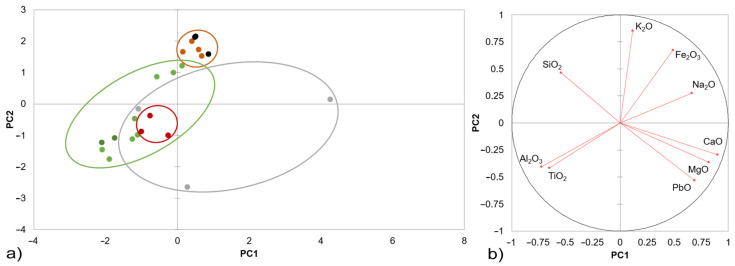
Principal component analysis biplot of the XRF data measured for the ceramic bodies of the studied archaeological fragments: (**a**) Romanesque floor tiles (brown dots), Gothic floor tiles (black dots), Renaissance stove tiles (green dots), large-format reliefs (dark-green dots), technical ceramics (red dots), and Modern Age utility ceramics and faiences (grey dots); (**b**) correlation circle with vectors of the investigated variables of the main oxides. The PC1 is the one that extracts the maximum variance, and PC2 is the one that extracts the maximum variance from what is left.

**Figure 3 materials-16-00375-f003:**
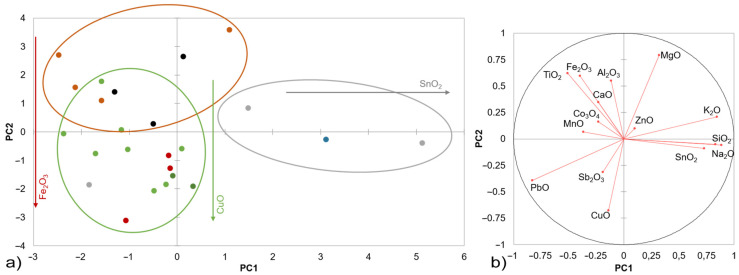
PCA biplot of the XRF data measured for the glazes of the studied archaeological fragments: (**a**) Romanesque floor tiles (brown dots), Gothic floor tiles (black dots), Renaissance stove tiles (green dots), large-format reliefs (dark-green dots), technical ceramics (red dots), Modern Age utility ceramics (grey dots), and blue onglaze decorations of faiences (blue dots); (**b**) correlation circle with vectors of the active variables. The PC1 is the one that extracts the maximum variance, and PC2 is the one that extracts the maximum variance from what is left.

**Figure 4 materials-16-00375-f004:**
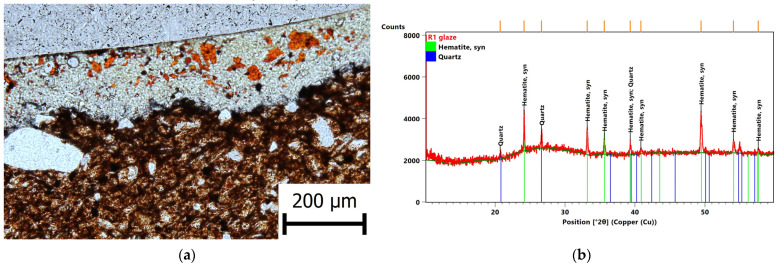
Hematite as colouring agent: (**a**) thin section image of a transparent glaze with hematite crystals and a corrosion product in the form of hydrated phosphates on the surface of the glaze; (**b**) XRD pattern of the glaze G3—the orange lines show positions of diffraction lines.

**Figure 5 materials-16-00375-f005:**
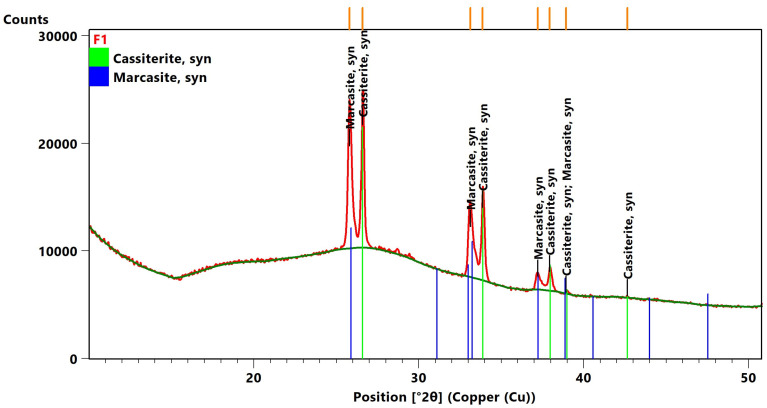
XRD pattern of the representative glaze F1 with cassiterite as the main crystalline phase of sample F1 white glaze—cassiterite accompanied by marcasite as a minor crystalline phase. The orange lines show positions of diffraction lines.

**Figure 6 materials-16-00375-f006:**
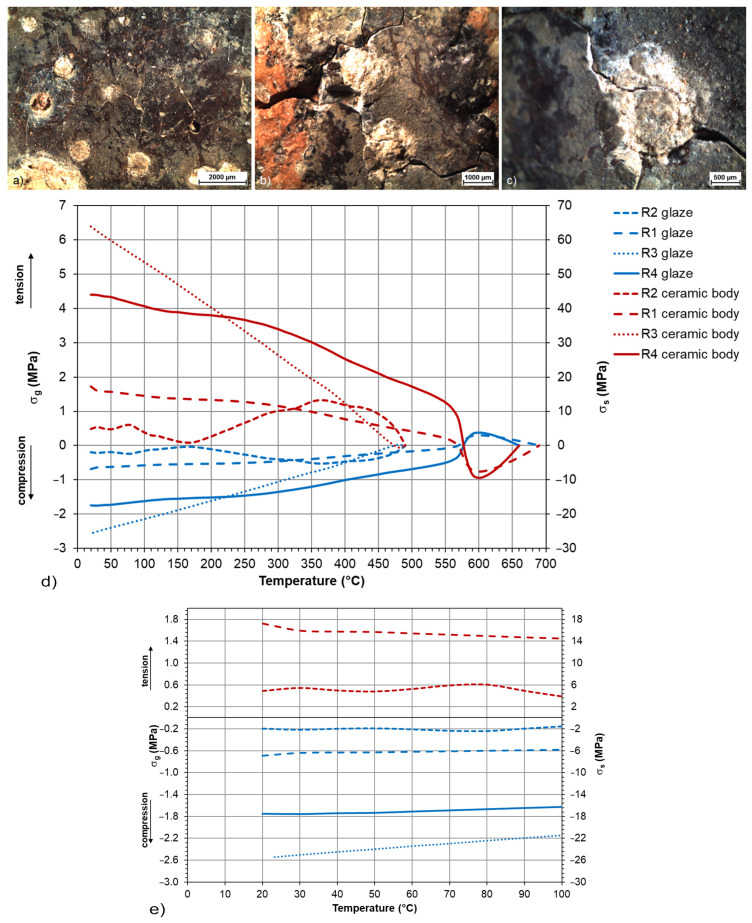
Images of degraded glazed surfaces of the Romanesque samples: (**a**) R1; (**b**) cracked surface of R4; (**c**) growth of corrosion products in the glaze of R4 due to deposit in soil with high quantity of phosphates; (**d**) stress relationships of the samples of the Romanesque glazed floor tile glazes measured by TMA, verified by the SciGlass calculations and the ceramic bodies measured by DIL; (**e**) the detail of the (**d**) in 85 % zoom.

**Figure 7 materials-16-00375-f007:**
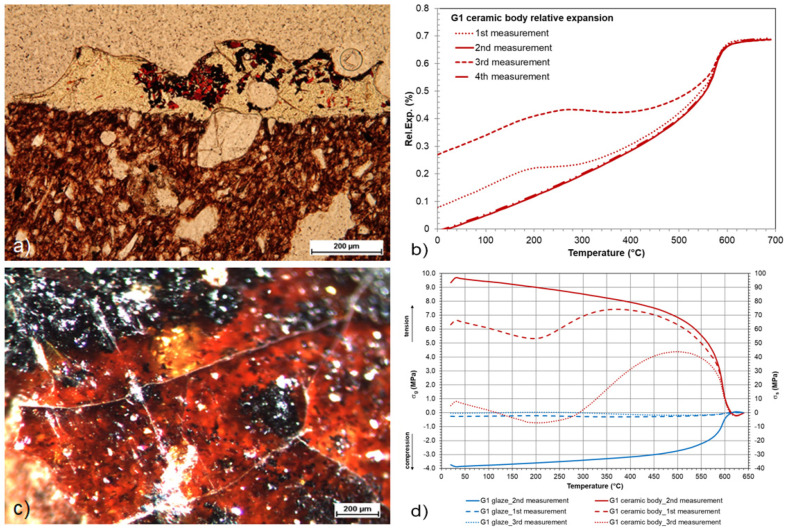
Sample of a Gothic tile: (**a**,**c**) stereomicroscope images of a degraded glaze; (**b**) relative expansions of the ceramic body measured four times; (**d**) stress states of the system calculated from each dilatometric measurement of the ceramic body.

**Figure 8 materials-16-00375-f008:**
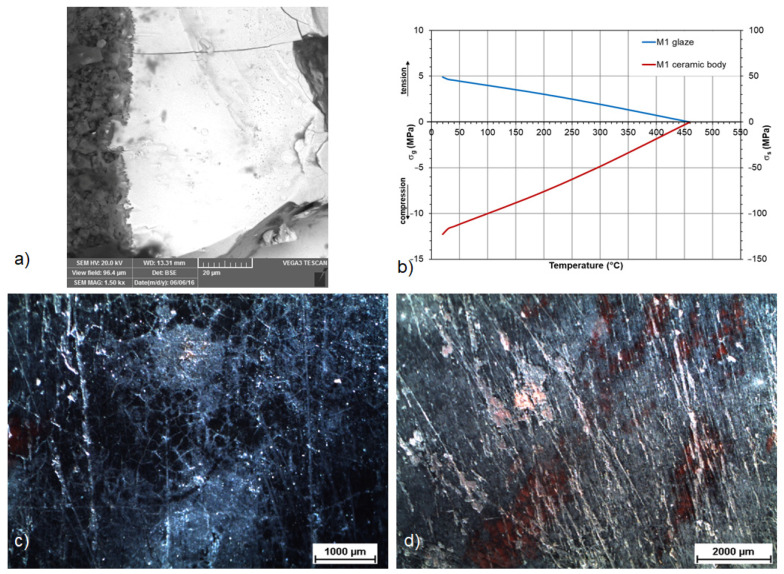
Tensile stress within the brown glaze of the pipkin M1 proves the poor glaze fit of the crazed glaze and accompanied by degradation mechanisms: (**a**) SEM image of the glaze layer; (**c**,**d**) stereomicroscope images of the crazed glaze with mechanical abrasions, crazing and pitting corrosion; (**b**) stress relationship of the M1 sample.

**Figure 9 materials-16-00375-f009:**
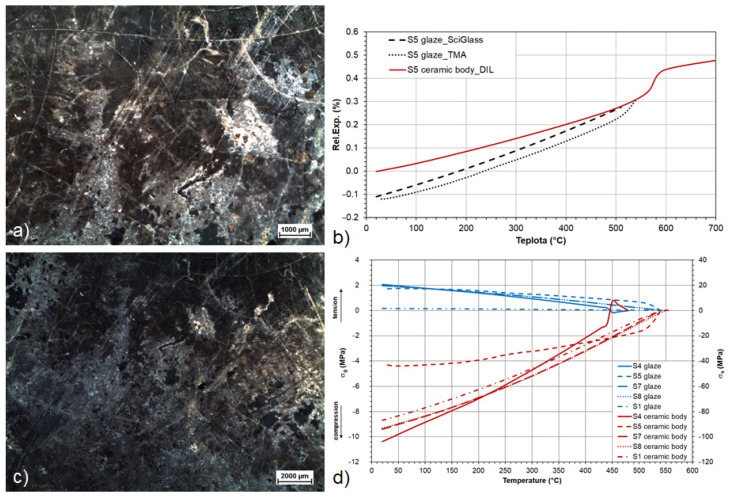
Tension behaviour of the stove tile glazes confirmed by the stress calculations: (**a**,**c**) crazed surface of the glazes of stove tiles S4 and S5 with pinhole corrosion; (**b**) calculated and measured relative expansion curves of stove tile S5; (**d**) stress relationships of stove tiles S1, S4, S5, S7, and S8.

**Figure 10 materials-16-00375-f010:**
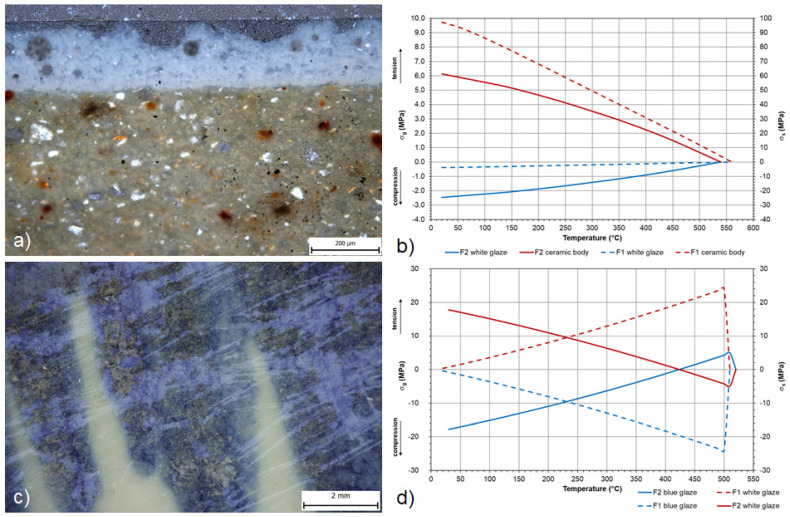
Flexion behaviour of the Renaissance Faience: (**a**) thin section image of sample F1; (**b**) stress relationships of the two-component system of white glazes and ceramic bodies; (**c**) stereomicroscope image of the abraded surfaces; (**d**) stress state of the two-component system of white and blue glaze.

**Figure 11 materials-16-00375-f011:**
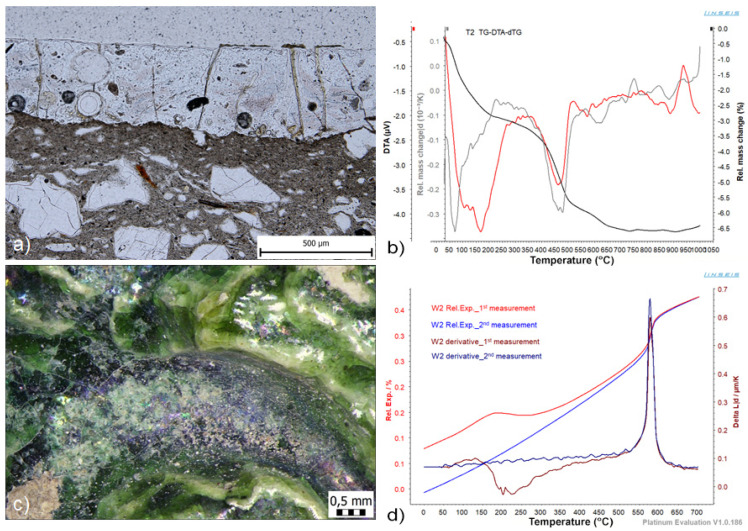
Moisture expansion of the ceramic bodies of the technical ceramics and large-format reliefs affected the glaze fit and flexion behaviour of the glaze-ceramic body systems T1, T2, T3, W1, and W2: (**a**) thin section image of the W2 sample; (**b**) TG-DTA-dTG curves of the ceramic body T2; (**c**) TG-DTA-dTG curves of the ceramic body T2; (**d**) relative expansion curves and numerical derivative as a function of the relative dilatation of the W2 ceramic body sample.

**Figure 12 materials-16-00375-f012:**
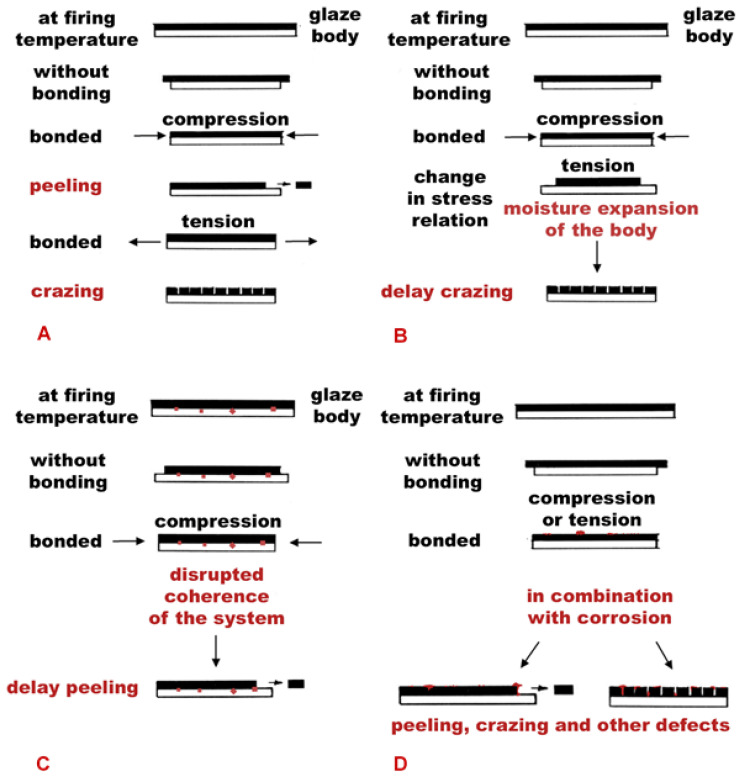
A scheme of the possible degradation and defect growth processes: (**A**) stress relationships and primary crazing or peeling of a glaze; (**B**) delayed (secondary) crazing of a glaze supported by moisture expansion of the body; (**C**) delayed (secondary) crazing of a glaze supported by moisture expansion of the body; (**D**) delayed (secondary) crazing of a glaze supported by moisture expansion of the body, red dots shows corrosion products.

**Table 1 materials-16-00375-t001:** Macro description of the investigated lead glazed fragments with representative optical images (The length of the scale is 10 cm).

Label	Photo Documentation	Location	Date	Description
R1	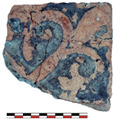	Lesser Town Square, Romanesque St. Wenceslas Rotunda on the premises of the Charles University building	11th to 12th century	Fragment of square tile with dark black-brown glaze
R2	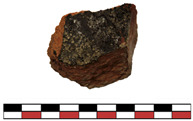			Fragment of square tile with dark black-brown glaze
R3	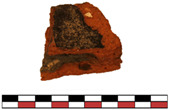			Fragment of square tile with dark black-brown glaze
R4	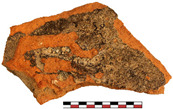			Fragment of hexagonal tile with dark black-brown glaze
G1	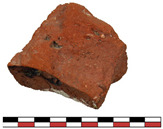	The western cloister of the Benedictine Arch-Abbey	13th century	Fragments from ceramic mosaic pavement with transparent and dark brown glazes
G2	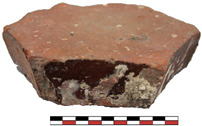			
G3	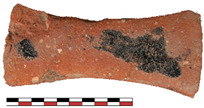			
S1	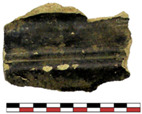	Prague Castle complex waste pit located in the Salm Palace	16th century	Pedestal corner of stove tile with brown glaze
S2	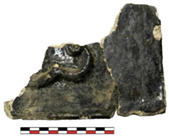			Fragment of stove tile with dark brown glaze
S3	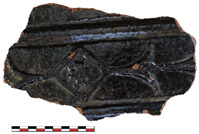			Fragment of stove tile with blue-brown glaze
S4	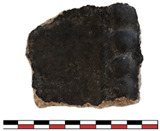			Fragment of stove tile with brown glaze
S5	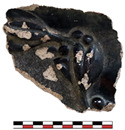			Fragment of stove tile with black-brown glaze
S6	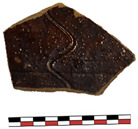			Fragment of stove tile with light brown glaze
S7	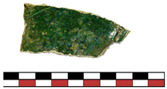	Prague Castle complex waste pit located in the Schwarzenberg Palace	17th century	Fragment of square stove tile with green glaze
S8	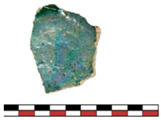		16th century	Fragment of rectangular stove tile with green glaze
W1	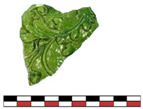	Prague Castle complex waste pit located in the Schwarzenberg Palace	16th century	Fragment of large-format wall tile with green glaze
W2	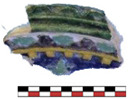	Prague Castle complex waste pit located in the Schwarzenberg Palace	16th century	Fragment of large-format wall tile with multi-coloured glazes
M1	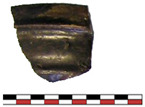	Prague Castle complex waste pit C	17th–18th century	Fragment of cooking pipkin with dark brown glaze
T1	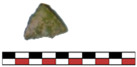	Prague Castle complex waste pit B	16th century	Fragment of distillation lid with green glaze
T2	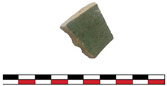			Fragment of technical bowl with green glaze
T3	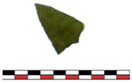			Fragment of technical bowl with green glaze
F1	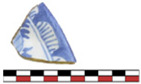	Prague Castle complex waste pit H and C	18th century	Fragment of a faience import plate with blue onglaze decoration
F2	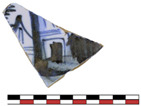			Fragment of a faience import plate with blue onglaze decoration

**Table 2 materials-16-00375-t002:** Calculated values of the stress σ and the total mismatch Δ at room temperature, the approximate set points T_n_ used for the calculations, and the summary of the main identified defects and corrosion products.

Label	Measuring Method	T_n_ (°C)	σ_body_ (MPa)	σ_glaze_ (MPa)	Total Mismatch Δ (MPa)	Identified Defects	Corrosion Products
**R1**	DIL/TMA	680	0.79	−0.05	0.84	delay peeling/mechanical abrasion	hydrated phosphates and carbonates
**R2**	DIL/calculation	490	5.43	−0.22	5.65	delay peeling/mechanical abrasion	hydrated phosphates and carbonates
**R3**	DIL/calculation	490	63.65	−2.54	66.19	speckling/mechanical abrasion	hydrated phosphates and carbonates
**R4**	DIL/TMA	660	44.00	−1.75	45.75	delay peeling/mechanical abrasion/ speckling	hydrated phosphates and carbonates (cerussite)
**G1**	DIL/calculation	640	96.92	−3.88	100.8	peeling/speckling/ delaycrazing	not present
**G2**	DIL/calculation	640	93.42	−3.74	97.16	peeling/speckling	not present
**G3**	DIL/calculation	640	86.65	−3.47	90.12	speckling/ delay crazing	not present
**S1**	DIL/calculation	560	−86.91	1.76	88.67	crazing/ salt efflorescence/speckling	hydrated phosphates
**S2**	DIL/calculation	580	−66.90	2.73	69.63	crazing/mechanical abrasion	not identified
**S3**	DIL/TMA	540	−68.17	3.01	71.18	crazing	not identified
**S4**	DIL/TMA	480	−100.73	2.07	102.80	crazing/corrosion pinholes/mechanical abrasion	hydrated phosphates and sulphates
**S5**	DIL/TMA	530	−43.19	1.73	44.92	crazing/corrosion pinholes	hydrated phosphates and sulphates
**S6**	DIL/calculation	500	−77.47	2.93	80.40	crazing	not present
**S7**	DIL/calculation	540	−94.05	1.97	96.02	crazing/ iridescence	hydrated phosphates
**S8**	DIL/calculation	550	−93.09	1.94	95.03	crazing iridescence/ corrosion pinholes	hydrated phosphates
**W1**	DIL/calculation	620	25.84	−0.36	26.20	delay crazing/salt efflorescence	not identified
**W2**	DIL/calculation	610	18.04	−0.26	18.30	delay crazing/iridescence	not identified
**M1**	DIL/calculation	460	−120.46	4.89	125.35	crazing/mechanical abrasion/corrosion pinholes	not identified
**F1**	DIL/calculation	560	90.09	−3.60	93.69	delay crazing/blisters/ mechanical abrasion	not identified
**F2**	DIL/calculation	540	61.35	−2.45	63.80	blisters/ mechanical abrasion	not identified
**T1**	DIL/calculation	630	18.04	−0.69	18.73	delay crazing/mechanical abrasion/ salt efflorescence	hydrated phosphates and sulphates
**T2**	DIL/TMA	620	17.20	−0.36	17.56	delay crazing/mechanical abrasion	hydrated phosphates and sulphates
**T3**	DIL/calculation	620	16.25	−0.65	16.90	delay crazing/mechanical abrasion	hydrated phosphates and sulphates

## Data Availability

Not applicable.

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
