# Peer review of "Degradation Processes of Medieval and Renaissance Glazed Ceramics"

_materials, 2022, doi:10.3390/ma16010375_

Round 1
Reviewer 1 Report
Dear authors, please find my comments in the attachment.

Author Response
Dear Reviewer,
thank you for your review and suggestions for improvement, the point-by-point response is uploaded in PDF below.
Yours faithfully.

Reviewer 2 Report
This paper uses a set of techniques to study Prague's different types of glazes. The test of the corrosion effects happened on the materials. Overall, the data is sufficient, and the conclusion is solid. However, the writing is not very scientific, and the figure editing skills need to be improved. Some of the figures may confuse the readers. After improving the figure and the writing, I think it is an interesting study worth publishing. Here are several comments on their figures:
1. In Figure 2 and Figure 3, what’s the difference between PC1 and PC2? Is it PC represent the principal component analysis? I think is better clarified in the figure caption.
2. For the XRD image, I think it is better to show the experiment data and the background curve more clearly (such as changing to another color and adding a legend for the data). And what does the yellow line represent in the XRD data? I think it is the experiment's peak position, but it is better to clarify it.
3. In the Figure 6 caption, what does the b represent?
4. In Figure 6, does the e represent the zoom-in value of Figure D? It is better to clarify it in the figure caption.
5. The scale bars formats are inconsistent in some figures; better keep them in the same format.
Author Response

(The authors gave the same response as above.)

Round 2
Reviewer 1 Report
The Authors implemented the suggested revisions